# DiBB: Distributing Black-Box Optimization

## Abstract

We present a novel framework for Distributing Black-Box Optimization (DiBB). DiBB can encapsulate any Black Box Optimization (BBO) method, making it of particular interest for scaling and distributing modern Evolution Strategies (ES), such as CMA-ES and its variants, which maintain a sampling covariance matrix throughout the run. Due to high algorithmic complexity however, such methods are unsuitable alone to address high-dimensional problems, e.g. for sophisticated Reinforcement Learning (RL) control. This limits the applicable methods to simpler ES, which trade off faster updates for lowered sample efficiency. DiBB overcomes this limitation by means of problem decomposition, leveraging expert knowledge in the problem structure such as a known topology for a neural network controller. This allows to distribute the workload across an arbitrary number of nodes in a cluster, while maintaining the feasibility of second order (covariance) learning on high-dimensional problems. The computational complexity per node is bounded by the (arbitrary) size of blocks of variables, which is *independent* of the problem size.

## 1 Introduction

Black Box Optimization (BBO) can be applied, by definition, to *any problem* independent of the specific application (Audet & Hare, 2017). In principle, this provides a method that is applicable to problems yet unsolved by the current state of the art. The most obvious catch lies in their computational requirements, which usually takes one of two forms: (i) simple black-box solvers suffers from high sample complexity (i.e., they are *data hungry*), in exchange for smaller computational requirements (both in memory and CPU time). On the other hand, (ii) sophisticated solvers such as modern Evolution Strategies (ES; Hansen & Ostermeier 2001; Wierstra et al. 2014) have a high internal computational cost per sample, in exchange for a wide set of properties (see Section 2) that make for significantly improved sample efficiency.

In the first case (simpler ES), the requirement for more samples can be mitigated—to some extent— through the embarrassingly parallel evaluation of a large population of candidate solutions. This approach however has its own limitations, as traversing a more complex fitness landscape often requires a disproportionate growth in the number of required samples as the dimensionality rises. In the second case however (complex ES), algorithms relying on covariance matrix adaptation (CMA; e.g. CMA-ES by Hansen 1996) have a quadratic complexity in the number of variables for processing a sample. This strictly limits their application (within a sensible time frame) to problems within the tens of thousands variables. This limit has restricted the applicability of these methods to rather simple scenarios and applications so far. As a result, most successful applications of ES to high-dimensional problems (e.g. training larger neural networks) rely on simpler ES, such as the type (i) described above.

Most of the per-sample complexity of type (ii) algorithms can be traced to the algorithm maintaining and updating the covariance information among the set of variables, which helps the sample efficiency of these algorithms (as in *convergence speed*) at the cost of longer run times (as in *wall-clock speed*), eventually becoming intractable as the number of variables grows. Several algorithms (Ros & Hansen, 2008; Schaul et al., 2011) thereby implement an assumption of *separability* between the variables, relinquishing covariance information altogether for a greatly enhanced wall-clock speed per sample. Unsurprisingly, this comes at a cost in terms of convergence speed: these algorithms can require several orders of magnitude more samples to find comparable solutions, depending on the problem structure (Hansen & Ostermeier, 2001).

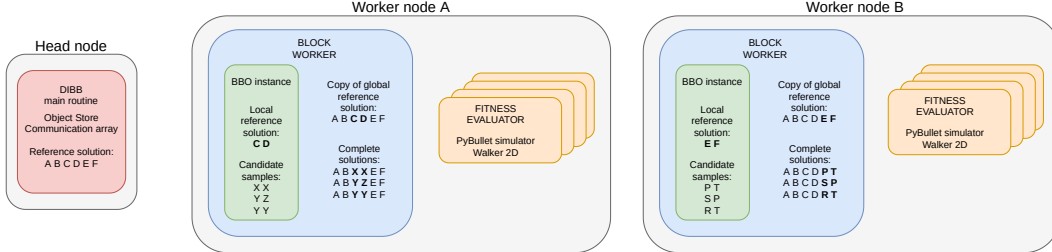

Figure 1: **Architecture of the DiBB framework.** This example shows an instantiation on a cluster with 3 nodes, one head and two workers.

The approach taken in this paper takes instead into account the fact that the correlation among variables is not uniform for most complex problems. Rather, certain groups of variables will display higher correlation between each other, with variables having lower correlation belonging instead into different groups. This induces two different assumptions on the variable set: groups having high intra-correlation support an assumption of *partial correlation*, while the low inter-group correlation displays the parallel argument of *partial separability*. We can represent this using a *block-diagonal* covariance matrix, which at the same time maintains full-covariance information within the variables belonging to one such group, and discarding the less useful (and often negligible) inter-group correlation altogether (Cuccu & Gomez, 2012). Such an approach allows leveraging covariance information where it is most efficient, while at the same time significantly reducing the memory footprint of the algorithm (based on the number and size of the groups), and even offering additional opportunities for parallelization and distributed computing. We will use the term *blocks* to refer to the groups of variable with high intra-correlation in the rest of the paper.

BBO algorithms are designed to ignore blocks, since they need to work in any unknowable (black-box) environment. Even in a gray-box setting, black-box algorithm engineering can be useful to ensure that no illegitimate assumptions are made in their design. On the other hand, we aim to profit from the added information. Real applications often have a gray-box character, exposing a certain level of expert knowledge about the correlation between variables. Let us take for example the common application of Neuroevolution (Stanley et al., 2019), where evolutionary algorithms (such as ES) are applied to learn the weights of a neural network. We know from the network's equation that the weights of the connections entering the same neuron (as opposed to entering different neurons) are by necessity highly correlated, as they are composed in a linear fashion inside the neuron prior to activation. The same reasoning can be made for the weights of connections entering neurons belonging to the same layer, versus neurons in different layers, because outputs of a layer are linearly combined in the next layer. This principle remains true as the network expands: in complex applications such as those requiring deep networks, the correlation between weights of connections in the early versus late layers will be by design much more limited than in the neuron or layer examples above[1].

Based on the above insights, this paper proposes a new framework for Distributed Black Box optimization, named **DiBB**, see Figure 1. We leverage the assumption of partial correlation by partitioning the variables set into highly intra-correlated blocks; then, switching the assumption to partial separability, we search each block with a different instance of the reference BBO algorithm, independently. This makes DiBB particularly suited for neuroevolution applications requiring larger/deep networks. In Section 2, we provide rigorous theoretical arguments for the sample efficiency of this approach.

From a practical perspective, this allows to search blocks asynchronously and even to distribute each BBO instance on different nodes in a cluster. Varying the partition of the variables enables the user to arbitrarily trade-off between wall-clock speed (smaller blocks) and sample efficiency (larger blocks), while distributing the BBO instances decouples the complexity from the number of variables, as long

---

[1]The fundamental assumption of partial separability across network layers however does not seem to be very well studied in the literature, despite a considerable body of work on neural network loss landscapes (Fort & Jastrzebski, 2019; Soltanolkotabi et al., 2018). This can be partially tracked to the limited availability of algorithms making use of partial correlation information.

as enough machines are available to run the BBO instances. A major challenge comes with solution evaluation, as the variables constituting a sample are practically scattered across a network. This is addressed in Section 3, by establishing a rather sparse communication scheme.

It is important to notice at this point that no assumption or claim has yet been made on the underlying BBO algorithm of choice—nor will be made. While this paper explores the applicability of DiBB in the context of modern, sophisticated ESs, DiBB can be applied to any BBO algorithm with minimal interfaces.

We analyze the performance of our framework on the classic COCO benchmark, both on the BBOB and BBOB large-scale suites, using the industry-standard CMA-ES as a reference, and as a block-level optimizer within DiBB. The sample complexity of our approach typically (and as expected) sits in between the full-covariance and the diagonal-covariance versions of CMA-ES, with few notable exceptions where maintaining extra covariance information is actually deceptive/misleading due to separability. Notably, on parallel hardware, DiBB is considerably faster in terms of wall clock time. To demonstrate the scalability of our algorithms, we present results on training a 20-layer neural network using 20 blocks on 20 machines, the largest neural network trained with neuroevolution to date, to the best of our knowledge.

## 1.1 CONTRIBUTIONS

Our key contributions are as follows:

- We provide a novel way of parallelizing an ES, going beyond the parallel evaluation of candidate solutions forming a population, by optimizing blocks of variables with multiple ES instances.
- We exploit the block structure to add covariance matrix information at low cost where it matters most, namely inside blocks of potentially highly correlated variables.
- The resulting DiBB framework is particularly well-suited for neuroevolution. However, it can be applied to any BBO method.
- We demonstrate that also large and deep networks can be trained efficiently with neuroevolution on highly parallel hardware.

## 2 A PRIMER ON EVOLUTION STRATEGIES

ES are direct search methods, which optimize a black-box objective function $f : \mathbb{R}^d \to \mathbb{R}$ by sampling candidate points from an adaptive Gaussian distribution. We briefly review the types of ES most relevant for our discussion. The classic variant is the (1+1)-ES (Rechenberg, 1973). Its central algorithmic mechanism is step size adaptation, i.e., its ability to actively adapt the standard deviation $\sigma > 0$ of its Gaussian sampling distribution $\mathcal{N}(m, \sigma^2 I)$ to the current needs. For about 20 years, CMA-ES (Hansen & Ostermeier, 2001) is the gold standard in ES research. Many variants exist, such as Natural Evolution Strategies (NES; Wierstra et al. 2014). Its most important mechanism going beyond "simple" step-size adaptive ES is covariance matrix adaptation (CMA), which means that not only the global step size $\sigma$, but also the full covariance matrix $C$ of the Gaussian $\mathcal{N}(m, \sigma^2 C)$ is adapted to the problem at hand.

CMA-ES is a powerful optimizer; however, it was not designed for high-dimensional applications with hundreds of thousands of variables. Its internal parameters are not tuned with such a regime in mind, and learning a full covariance matrix with $\frac{d(d+1)}{2}$ parameters is inherently slow. Such problems are best addressed by restricting $C$ to a diagonal matrix (Ros & Hansen, 2008), to a diagonal plus a low-rank matrix (Loshchilov, 2014; Akimoto et al., 2014; Loshchilov et al., 2018), or to a block-diagonal matrix (Cuccu & Gomez, 2012). The number of parameters of the covariance matrix can hence be chosen rather flexibly in the range $d$ to $\frac{d(d+1)}{2}$.

## 2.1 ES FOR NEUROEVOLUTION

The application of ES to machine learning problems and to RL in particular has a long history (Igel, 2003; Heidrich-Meisner & Igel, 2009). In 2017, the work of Salimans et al. (2017) sparked a renewed

interest in ES by showcasing how to successfully exploit the embarrassing parallel nature of objective function evaluations in populations-based algorithms, considerably speeding-up the learning process. This triggered a large body of work on neuroevolution based on ES, see e.g. Plappert et al. (2017); Ha & Schmidhuber (2018); Chrabaszcz et al. (2018); Stanley et al. (2019) and references therein.

## 2.2 Convergence Rates and Computational Complexity

Due to Taylor's theorem, local optima of $\mathcal{C}^2$ functions in $d$-dimensional space are well approximated (up to $\mathcal{O}(\|x - x^*\|^3)$) by convex quadratic functions $f(x) = \frac{1}{2}(x - x^*)^T H (x - x^*)$. The computational complexity of solving this problem with an ES to a fixed target precision $\varepsilon > 0$ is of the form $\mathcal{O}(d \cdot \kappa(H) \cdot \log(1/\varepsilon))$, where $\kappa(H)$ denotes the condition number of the Hessian $H$ (Jägersküpper, 2006; Hansen et al., 2015). Hence, a step-size adaptive ES achieves linear convergence with rate $\mathcal{O}(1/(d \cdot \kappa(H)))$.

The linear dependency on $d$ is optimal for comparison-based optimization (Fournier & Teytaud, 2011), but the dependency on $H$ is sub-optimal. The advantage of maintaining covariance information is that the factor $\kappa(H)$ is improved to $\kappa(HC^*)$, where $C^*$ denotes the optimal covariance matrix available to the ES. When approaching $C^* = H^{-1}$, methods maintaining full covariance achieve the optimal value $\kappa(HC^*) = 1$. In effect, as expected from a pseudo second order method, the convergence rate is *independent* of $H$. A diagonal covariance matrix $C^*$ acts as a diagonal pre-conditioner, with varying effectiveness depending on the problem. Obviously, the block-diagonal and the low-rank cases are in-between.

The OpenAI-ES (Salimans et al., 2017) features neither step size adaptation nor covariance adaptation. Based on the NES framework of Wierstra et al. (2014), it leverages the ability of ES to estimate the natural gradient of $f$ from samples, and then applies the ADAM optimizer on top. In effect, this is roughly comparable to using diagonal $C$.

CMA has a price in terms of *algorithm internal* complexity, and in addition the adaptation process is slow in terms of sample complexity. The above convergence rates measure time in terms of objective function evaluations (sample complexity). When scaling up CMA to high dimensions, we need to take the following concepts into consideration. Algorithm internal complexity refers to the required (amortized) number of operations needed for creating a sample and for updating the internal state—the covariance matrix in particular. Regarding sample complexity, we distinguish between the number of samples needed to learn the covariance matrix, and the number of samples needed to solve the problem.

Learning $C$ with up to $\Theta(d^2)$ parameters is sample-inefficient. For example, a (small) network with $d = 10^4$ weights results in $\frac{d(d+1)}{2} \approx 5 \cdot 10^7$ parameters of the covariance matrix. Learning these takes hundreds of millions of samples, each of which can be a full RL episode. This is too slow for being useful. For larger $d$, even the storage of the full covariance matrix $C$ becomes prohibitive. Furthermore, performing computations with $C$ scales at least linear with the number of its parameters, which amounts to an internal complexity of $\Omega(d^2)$ for full CMA. Since network evaluation scales linearly with the number of weights $d$, CMA quickly becomes the computational bottleneck. Therefore, a different trade-off between fast convergence and internal complexity is needed, which can be realized for example with block-diagonal and low-rank structures, and combinations thereof.

## 2.3 Implications for Neural Network Training

Diagonal, block-diagonal, and low-rank schemes successfully lower both internal and sample costs of CMA significantly. Therefore, they are key to the application of modern ES with CMA to RL.

In neural network training, weight spaces are often extremely high-dimensional. However, they also come with a canonical structure, induced by the network topology. We can expect weights belonging to different layers (and even weights belonging to different neurons within the same layer) to be less correlated than weights within a layer. Hence, a block-diagonal covariance structure with one block per layer (or per neuron) is a natural choice. It should be noted that there exist approaches for identifying a problem decomposition automatically (Omidvar et al., 2013), if needed.

If $H$ has a block structure ($f$ is separable), then *sequentially* optimizing all blocks in isolation is as fast as optimizing the full problem. This is a direct consequence of the $\mathcal{O}(d)$ scaling of the sample complexity discussed above, and lead to the following insight:

> Solving $b$ independent sub-problems with $k = d/b$ variables each *in parallel* results in a $b$-fold speed-up over solving the full problem with $d$ variables.

It is understood that this comes at a cost if the separability assumption is violated. However, the block structure offers a further benefit:

> Solving $b$ independent sub-problems with $k = d/b$ variables each with an ES featuring CMA results in $\mathcal{O}(k^2)$ sample and internal complexity for covariance learning, in contrast to $\mathcal{O}(d^2)$ for full CMA. If the number of blocks $b$ scales linearly with the problem size $d$, or $k \in \mathcal{O}(1)$, then CMA becomes feasible for arbitrarily large problems.

These two insights offers a novel route towards highly parallel and at the same time more sample-efficient neuroevolution strategies. In addition to the embarrassingly parallel evaluation of a population of candidate points, multiple blocks can be optimized in parallel. Given enough cores, higher parallelism results in a $b$-fold speed-up. As an additional benefit, CMA can be applied within each block of variables. Provided that the problem has an (approximately) separable structure, CMA results in improved sample efficiency, yielding a further speed-up.

## 3   METHOD

The DiBB framework is originally inspired by Block-Diagonal Natural Evolution Strategies (BD-NES; Cuccu & Gomez 2012), but extending the original design to be applicable to any BBO algorithm without restrictions, providing at the same time a parallel and distributed implementation with limited overhead. Here is a short summary of how to use DiBB:

1. The user defines a partition of the parameters, typically based on expert knowledge of the application, e.g. for neuroevolution consider one block per (weights of connections entering a) layer or one block per neuron.
2. Launching the method on a *head node* spawns the run control routine plus the object store that maintains shared data. This follows typical best practices in distributed processing (Dean & Ghemawat, 2008).
3. The control then launches one *Block Worker* (BW) for each parameter block on the available machines. The BW encapsulates the actual BBO algorithm. Each BW runs fully asynchronously from the others, though contemporary. The BWs exchange information by uploading to the head node the state of their search after each update (i.e. a *generation* in ES).
4. In our implementation, each BW can spawn a pool of *Fitness Evaluators* (FE) on the same machine. These are used to manage limited computational resources, such as available CPUs, which becomes critical when a nontrivial objective function needs access to a significant portion of the available resources, such as the case with external software (e.g. physics simulations of control tasks).
5. Alternatively, the user can request to evaluate trivial optimization functions on the BW directly (either sequentially or with multi-threading), which is useful when the overhead of maintaining a discrete pool of evaluators would be significant.

The Block Workers communicate with the head node in each generation, while generation cycles are defined asynchronously by each BW. Consider a problem with $d$ variables $x_1, \ldots, x_d$, and a BW optimizing $b$ variables $x_a, \ldots, x_{a+b-1}$, denoted as the vector $x_B$ for short. The head node maintains a *reference solution* $\bar{x} \in \mathbb{R}^d$, which fulfills a two-fold purpose: it serves as an anytime estimate of the state of the search (optimum), and it provides a unifying context to the BWs and the FEs.

Intuitively, each BW is only aware of the variables in one block, and can only generate samples for the corresponding variables. These incomplete samples need to be scored on the task to obtain the gradient for solution improvement. The problem here is that the task expects a complete solution to be evaluated: in our recurring example of neuroevolution, the sample could correspond to the weights for a neuron, or layer, while only full networks can be evaluated on the task.

We address this issue by leveraging once again our assumption of partial separability. After our hypothesis of the correlation across blocks being negligible, we can evaluate each block in isolation by inserting it in the context of the reference solution, obtained from the head node. Hypothesize for a moment that there is no correlation inter-block—we address below the validity of this statement. In this case, each layer can be scored fairly by constructing a full reference network, then swapping the corresponding weights in the target layer for the block sample, and finally evaluating the resulting complete network on the task. Different independent samples from a same block would receive fair evaluation in this fashion as long as they are evaluated on the same reference network. This is in fact constructed by assembling the per-block, partial *reference solutions* of each of the BBO instances running on each block, and is updated in the head node by each block instance after each generation.

More formally: at the start of each generation, the BW receives the current reference solution $\bar{x}_1, \ldots, \bar{x}_d$ from the head node.[2] The block-level ES samples a population of candidate solutions $y^1, \ldots y^\lambda \in \mathbb{R}^b$. If $b$ is small, then sampling from a Gaussian with full covariance matrix $\mathcal{N}(m, \sigma^2 C)$ is feasible. The $b$-dimensions points are injected into the reference solution by constructing the $d$-dimensional vectors $x^1, \ldots x^\lambda \in \mathbb{R}^d$ according to the following rule:

$$x_i^j = \begin{cases} y_{i-a}^j & \text{if } a \leq i \leq a + b \\ \bar{x}_i & \text{otherwise} \end{cases}$$

The vectors $x^1, \ldots, x^\lambda$ are passed to the (thread or node) pool of Fitness Evaluators. Once all objective function values are computed, the ES updates its internal state, which contains its sampling mean $m$ and optionally further parameters like step size, evolution paths, and covariance matrix. Finally, it sends the updated mean $m$ back to the head node, which incorporates it into its reference solution by overwriting $x_B$ with $m$.

The last issue remaining is that of course we cannot expect the weights entering different neurons or layers to be entirely separable; after all, they belong to the same network and they are thereby all contributing to the final output in its equation. This results in a problem of *moving target*, where the score of a block sample depends on the global state of the search (in the form of the current complete reference solution), which changes constantly as every BBO instance asynchronously sends an update to the global reference state. Empirically we verify that the impact on the algorithm is only marginal: after all, our hypothesis is relatively much stricter than in fully-separable implementations. This is further mitigated when comparing with full-covariance algorithms by the fact that the latter will still need to learn the lower covariance between lowly-correlated variables, which in turn effectively lowers the sample efficiency of the theoretically superior algorithm in most complex real-world scenario. In practice, we have seen no measurable advantage or disadvantage on either approach from this perspective.

The proposed setup directly reflects the added level of parallelism, compared to a standard ES. Traditional implementations usually restrict parallelism to maintaining a pool of evaluators to speed up the evaluation of independent samples using multiple cores. In our setup, a second level of parallelism is established in terms of the Block Workers, which operate independently but for sparse communication with the head node.

## 4 EXPERIMENTS

This section describes the setup used to empirically assess the performance of DiBB. With our experiments, we aim to address the following research questions:

Q1: How well does the block-diagonal approach work, compared to diagonal and full CMA?
Q2: How does DiBB scale to a large number of machines and cores?
Q3: Is DiBB well-suited for neuroevolution applications?

We assess the first two questions on the standard COmparing Continuous Optimizers (COCO) Black Box Optimization Benchmark (BBOB), using both the standard and the large-scale benchmark suites. For answering the third question, we showcase a neuroevolution application in the challenging OpenAI Gym 2D Walker environment.

---

[2]It would suffice to send $\bar{x}_1, \ldots, \bar{x}_{a-1}, \bar{x}_{a+b}, \ldots, \bar{x}_d$ to the BW, but the difference is usually negligible.

**Cluster setup.** We tested DiBB on a variety of hardware solutions. The COCO-BBOB experiments were run on a cluster of 24 low-performance machines, each sporting an Intel$^{(R)}$ Core$^{(TM)}$ i7-2600 CPU @ 3.40GHz (4 cores/8 threads each), and 32 GB of RAM. As one can expect, these old machines are far from state-of-the-art performance, which leaves a significant margin of improvement for the timings presented in our results.

**Reference implementation.** Our experiments use our implementation of DiBB written in Python and leveraging the Ray distributed computation library[3]. The code is released open source on GitHub[4], and available through Pypi[5] for ease of adoption and extension. The cluster setup is simplified by the included managing scripts. Running on a single machine with a single block and no Fitness Evaluators roughly corresponds to running the underlying BBO algorithm alone (plus overhead). Spawning multiple BWs and FEs automatically scales to the available resources. Defining a set of IPs in the managing script allows to easily distribute the computation.

## 4.1 COCO BBOB

COCO provides multiple suites covering a broad range of test cases. The experiments presented in this work are based on the BBOB and the BBOB large-scale suites.

**BBOB.** The BBOB suite comes with 24 noise-free real-parameter single-objective benchmark functions in dimensions $d \in \{2, 3, 5, 10, 20, 40\}$ (Hansen et al., 2009). The functions are divided into five groups: separable functions (f1-f5), moderately conditioned functions (f6-f9), ill-conditioned functions (f10-f14), multi-modal functions (f15-f19), and weakly structured multi-modal functions (f20-f24). We expect the performance of DiBB to vary accordingly to our assumption of partial separability.

**BBOB large-scale suite.** The BBOB large-scale suite contains the same 24 same functions which are found in the BBOB suite, but scaling the available number of dimensions to $d \in \{20, 40, 80, 160, 320, 640\}$. This suite however introduces heuristics to decrease the computational cost of a selection of functions in a large-scale setting. We refer the interested reader to Elhara et al. (2019) for further details.

The BBOB suite provides a broad range of problems representative of many use-cases in the spectrum of Black-Box Optimization. Particularly, and by design, several edge cases are present that are unlikely to be encountered in real applications but provide compelling insights on the performance and applicability of the tested algorithm. Particularly, BBOB problems are designed to be solved to high precision to test for scale invariance; therefore $d$ is not in the thousands. Scalability of $d$, and the distinction between separable and non-separable problem classes, allows us to systematically evaluate the effect of DiBB's block structure on its performance. COCO greatly facilitates this process by providing publicly available performance data for many SOTA algorithms.[6]

We test the scaling of DiBB's performance as we vary the problem dimensions and block size, from the opposing perspectives of sample efficiency (i.e. convergence speed) and run time (i.e. wall-clock speed). To this end, we ran the following experiments:

1. Constant number of blocks, increasing $d$ and block size, on the BBOB suite (`bbob_fnb`)
2. Constant block size, increasing $d$ and number of blocks on the BBOB suite (`bbob_fbs`)
3. Constant number of blocks, increasing $d$ and block size on the large-scale suite (`bbob_ls_fnb`)
4. Constant block size, increasing $d$ and number of blocks on the large-scale suite (`bbob_ls_fbs`)

Details of the experimental setup are found in Appendix A.1.

---

[3]`https://www.ray.io/` — a Python framework for distributed computing

[4]Link anonymized.

[5]Link anonymized.

[6]`https://numbbo.github.io/data-archive/bbob/`

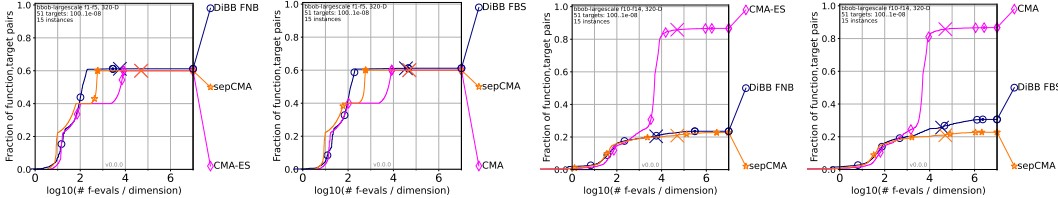

Figure 2: ECDF plots of the performance of DiBB, separable and full CMA-ES in dimenion 320. From left to right: f1-f5 with Exp 1/Exp 3, Exp 2/Exp 4, then f10-f14 with Exp 1/Exp 3, Exp 2/Exp 4.

We chose to focus on the group of separable (f1-f5) and ill-conditioned functions (f10-f14). The first group contains problems that can be solved easily by DiBB, while the second group is extremely hard since it strongly violates the separability assumption.

Prototypical results of the COCO/BBOB experiments are found in figure 2 (full results in the appendix). The ECDF plots show the fraction of reached (precision) targets over dimension-normalized time on a log-scale, so "higher is better".

**Sample Efficiency**  On fully separable problems f1-f5, all algorithms perform roughly the same in terms of sample complexity. In other words, covariance terms can be dropped at no cost, and blocks can be optimized independently. This confirms the theoretical predictions from Section 2. Interestingly, in some cases DiBB (slightly) outperforms full CMA-ES, since off-diagonal covariance terms can be detrimental when not needed.

For the ill-conditioned non-separable problems f10-f14, the performance of DiBB is close to CMA-ES with diagonal covariance matrix and far worse than full CMA-ES, since the unmodeled terms with (unrealistically high) correlations extremely close to $\pm 1$ dominate performance. Yet, unsurprisingly, larger blocks result in better sample complexity. This finding confirms the theoretical prediction that the assumption of separability is crucial.

Taken together, the two results imply that DiBB applied to a problem with block structure greatly outperforms an ES with diagonal covariance matrix by being more sample-efficient within each block, while full CMA offers no further advantage. This answers question Q1.

**Timings**  Most of the experiments were run on the low-performance cluster described above. For 3 of the experiments however we tested the flexibility of DiBB by switching to a new cluster of nodes with Intel(R) Xeon(R) CPU E5-2620 v4 @ 2.10GHz processors (32 cores in total) and 128 GB RAM. These were the 5d BBOB suite with one block (thus running on a single machine), the 40d BBOB suite with 2 blocks (running on 3 machines: 1 head and 2 for the Block Workers), and the 40d BBOB large-scale suite with one block (again single-machine run).

| Number of dimensions | Number of blocks | Block size | Duration | Number of dimensions | Number of blocks | Block size | Duration |
|---|---|---|---|---|---|---|---|
| | | | | 40 | 1 | 40 | 25h 13m 47s |
| 80 | 16 | 5 | 02h 47m 53s | 80 | 2 | 40 | 43h 26m 57s |
| 160 | 16 | 10 | 04h 11m 15s | 160 | 4 | 40 | 40h 21m 01s |
| 320 | 16 | 20 | 07h 01m 38s | 320 | 8 | 40 | 44h 53m 29s |
| 640 | 16 | 40 | 12h 16m 02s | 640 | 16 | 40 | 61h 20m 11s |

Table 1: Timing data for experiment 3 on the BBOB large-scale suite with a fixed number of blocks (left) and experiment 4 with a fixed block size (right, $5\times$ larger budget).

Table 1 compares the runtimes of DiBB with different numbers of blocks, block sizes, and dimensions on the BBOB large-scale suite. Several effects come together: in larger dimensions, function evaluation time and communication overhead grow linearly. Due to the fixed budget multiplier used in COCO, the overall function evaluation budget also grows linearly. On the other hand, CMA's computational effort grows quadratically in the block size. We clearly observe that the runtime grows far more benign when using a fixed block size, which indicates that CMA overhead indeed quickly

becomes the dominating term. Hence, the block size should be kept tightly under control. This answers question Q2.

## 4.2 PyBullet Walker 2D

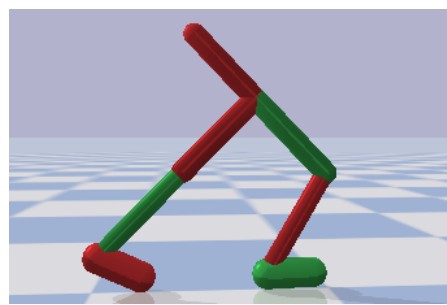

Figure 3: **The Walker 2D robot** from the `Walker2DBulletEnv-v0` environment in OpenAI Gym. The goal is to make the robot walk as far as possible without falling. This sophisticated control task provides a significant challenge in reinforcement learning control, especially in its PyBullet implementation. We tackle it using DiBB wrapping LM-MA-ES to evolve a neural network policy controller totaling 198 neurons and 11590 weights in direct encoding.

Although this paper introduces DiBB as a generic BBO framework, the original motivation behind its design is rooted in scaling neuroevolution to large network models. Therefore we present our results on a complex reinforcement learning control task: the Walker 2D gym environment from PyBullet (Coumans & Bai, 2016–2021).

This task, running on the PyBullet physics engine, is about low-level control of a fairly basic 2D robot (see Figure 3). The goal of the task is to learn a gait that allows the robot to reach the farthest possible distance in a limited time and without toppling. The 22 dimensions of the observations correspond to a broad array of sensors including positional and angular speed of the joints, height above the ground, speed along the 3 axes (plus roll, pitch and yaw), and more. The action space is composed of 6 (normalized) torque control signals, which are sent to the motors in each joint.

The policy network is feed-forward and fully connected, including 2 hidden layers of sizes [128, 64], and using ReLU activation on all neurons. The output layer is composed of 6 neurons with tanh activation to match the normalized range of motor commands.

The network is trained with DiBB wrapping LM-MA-ES (Loshchilov et al., 2018), and reaches a score of **1126** (average score over 100 runs, maximal episode length 1000) in 25 hours running on a cluster of 4 machines (3 of which have 8 Intel(R) Core(TM) i7-3770 CPU @ 3.40GHz processors, the fourth one having 32 Intel(R) Xeon(R) CPU E5-2620 v4 @ 2.10GHz processors). As a comparison we used Random Weight Guessing, generating 1000 individuals, which reached a score of 42.These are the first results on this benchmark using neuroevolution to the authors' knowledge, due to the limited availability of sophisticated evolutionary methods scaling to that many dimensions. This answers question Q3.

## 5 Conclusions

We present a new framework that wraps any Black-Box Optimization (BBO) algorithm into a new partially-separable variant. The variable set is partitioned into *blocks* with high intra-correlation but low inter-correlation, based on expert knowledge on the problem structure that is commonly available in real-world applications. Each block is searched independently by a separate instance of the wrapped BBO algorithm, which allows to parallelize and distribute its computation across any number of available machines. We name it DiBB for Distributing Black-Box optimization.

Its main advantage lies in bounding the overall computational complexity not in the total number of variables, but in the (arbitrary, user-defined) size of the largest block. The algorithm complexity scales constantly with the number of blocks as long as more machines are available, but for a limited communication overhead. Varying the number and size of the blocks allows the user to trade off convergence speed (larger blocks giving more awareness of the relationship between variables) and wall-clock speed (smaller blocks make for faster updates). DiBB is particularly well suite for the purpose of scaling neuroevolution to larger network models, with (weights of connections entering) neurons and layers being prime candidate for block grouping. **Future work** includes exploring the dynamics of different BBO algorithms, and scaling to larger models—potentially non-differentiable, as smoothness is not a requirement.

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

## A  APPENDIX

### A.1  DETAILS OF THE COCO/BBOB EXPERIMENTS

For experiments 1 and 2, a budget multiplier of $10^5$ was chosen and the IPOP restart mechanism was used in accordance with Hansen (2019). To compare not only against CMA-ES but also the separable version, we also ran sep-CMA-ES on the bbob suite with a budget multiplier of $10^4$.

To compare with CMA-ES and sep-CMA-ES, we ran experiments 3 and 4 without IPOP and we sampled the initial candidate solutions uniformly at random from $[-4, 4]^d$ the same way as Varelas (2019). Experiment 3 uses a budget multiplier of $10^4$, and experiment 4 uses a budget multiplier of $5 \cdot 10^4$. The initial step size `sigma0` was set to 2 in all of the experiments.

The data for CMA-ES and sep-CMA-ES can be downloaded via `cocopp` or using the following links respectively: `https://numbbo.github.io/gforge/data-archive/bbob-largescale/2019/CMA_Varelas_largescale.tgz` and `https://numbbo.github.io/gforge/data-archive/bbob-largescale/2019/sepCMA_Varelas_largescale.tgz`.

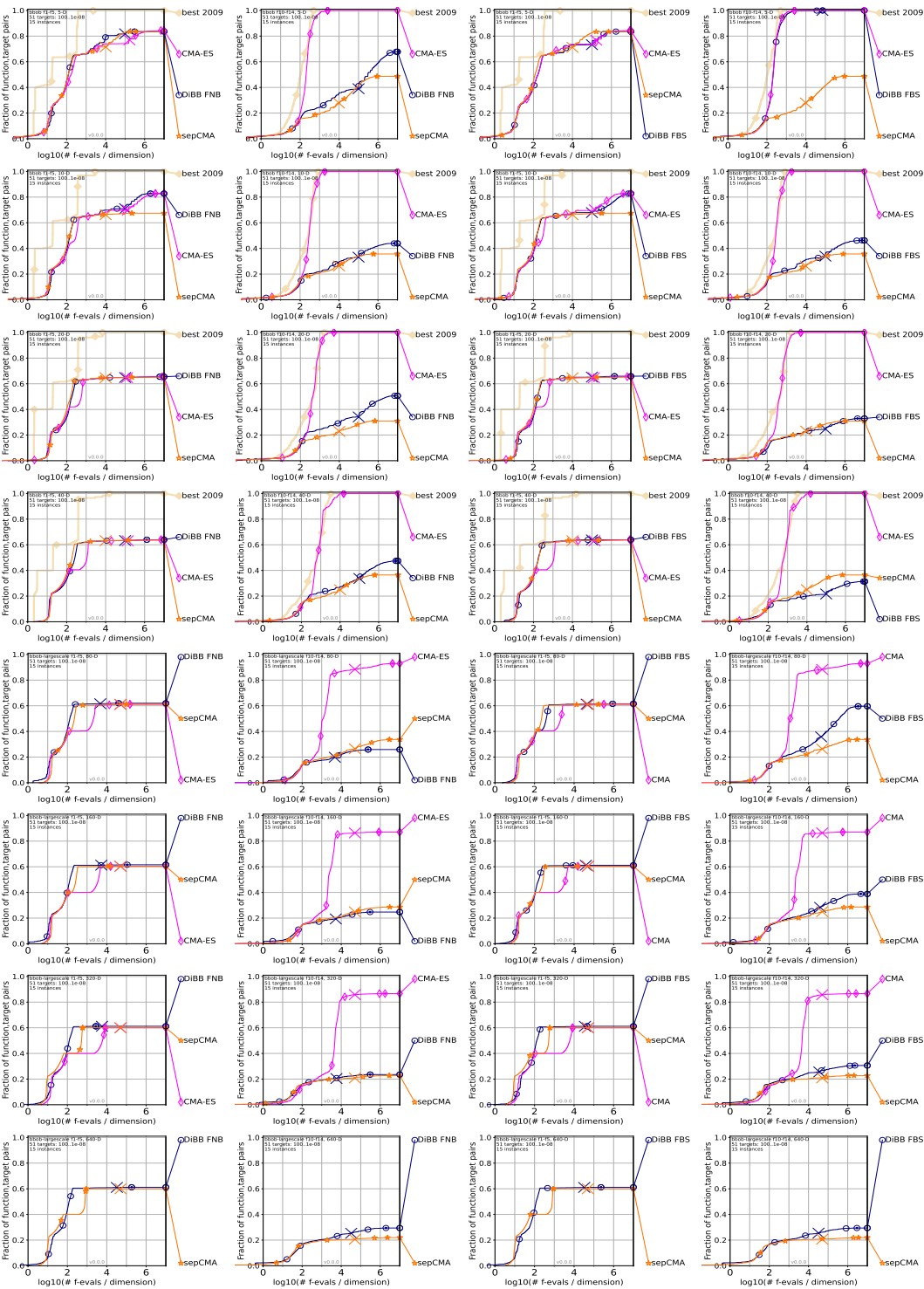

Figure 4: BBOB-COCO results of DiBB, separable and full CMA-ES. Rows correspond to dimensions 5, 10, 20, 40, 80, 160, 320, and 640. Columns correspond to Exp 1/Exp 2 on separable functions, ill-conditioned functions, as well as Exp 3/Exp 4 on separable and ill-conditioned functions.

| **BBOB** | $d = 5$ | $d = 10$ | $d = 20$ | $d = 40$ |
| --- | --- | --- | --- | --- |
| **Exp. 1** | 2 / 2,3 / 250K | 2 / 5 / 500K | 2 / 10 / 1M | 2 / 20 / 2M |
| **Exp. 2** | 1 / 5 / 500K | 2 / 5 / 500K | 4 / 5 / 500K | 8 / 5 / 500K |
| **large-scale** | $d = 80$ | $d = 160$ | $d = 320$ | $d = 640$ |
| **Exp. 3** | 16 / 5 / 250K | 16 / 10 / 500K | 16 / 20 / 1M | 16 / 40 / 2M |
| **Exp. 4** | 2 / 40 / 400K | 4 / 40 / 400K | 8 / 40 / 400K | 16 / 40 / 400K |

Table 2: Experimental setup on the BBOB and the BBOB large-scale suites, in the format "number of blocks / block size / # objective function evaluations per machine". Experiments 1 and 3 scale the block size (and correspondingly, the evaluation budget per compute node), while experiments 2 and 4 scale the number of blocks.

