# OpenReview forum: "DiBB: Distributing Black-Box Optimization"
_ICLR.cc/2022/Conference — ICLR 2022 Submitted_

### Official Review · Reviewer_CxhD · 2021-10-21

**Correctness:** 3
**Technical Novelty And Significance:** 2
**Empirical Novelty And Significance:** 2
**Recommendation:** 3
**Confidence:** 4

**Main Review:**

Strengths:
* Paper is clearly written and I was able to understand the technical concepts behind it.
* Method is highly parallelizable and is scalable across multiple different machines.
* Comprehensive results over BBOB functions show the speed scaling as well as which types of functions DiBB shines.


Weaknesses:
* Motivation for the Mujoco (actually, only Walker2d) benchmark. Applying LM-MA-ES on Walker2D (with a relatively huge policy network) is a bit artificial, since it's already solvable by a significantly smaller (even linear) policy with purely ARS/ES (see [1]). Figure 4 in [2] also showed that LM-MA-ES performs significantly worse than ES, so it's even more unclear why we need a Hessian-based method here. I would suggest to the authors to use a benchmark where CMA/Hessian methods are actually needed, over ES/ARS. Outside of the specific paper, it is generally a standard to also try methods on all standard Mujoco benchmarks (e.g. HalfCheetah, Hopper, Swimmer, etc.) and not just Walker2d.

* Theoretical foundations are still quite lacking, even though the paper discusses convergence rates (ex: 2.2, 2.3). In Section 2.3, the paper discusses that "weights belonging to different layers (and even weights belonging to different neurons within the same layer) to be less correlated than weights within a layer." This is somewhat of a controversial statement and needs to be quantified carefully. Can you provide any form of justifications for this claim? A simple and cheap way for instance, is to measure this correlation matrix explicitly (see ex: Figure 1 in [2]) for an example problem.

* The method requires the user to know in advance which parameters have "high correlations" between each other rather than potentially learning this separation. This makes it fairly impractical for many scenarios and fairly difficult for widespread adoption. I would suggest to the authors to propose a mechanism to automate this process, as it seems like it may be a huge issue.

[1] Simple random search of static linear policies is competitive for reinforcement learning (NeurIPS 2018)

[2] From Complexity to Simplicity: Adaptive ES-Active Subspaces for Blackbox Optimization (NeurIPS 2019)

**Summary Of The Paper:**

This paper proposes Distributed Black Box optimization (DiBB), which involves using disjoint distributed pipelines to perform CMA/Hessian-based updates, over functions with assumed separability in terms of parameters. The method is explained in detail, and experiments are performed on BBOB functions (with varying dimensions), along with Mujoco Walker2d with a relatively large policy architecture (10K+ parameters). Other ablations such as wall-clock time and varying block sizes are also experimentally presented.

**Summary Of The Review:**

The paper seems like it has promise in terms of the core idea (i.e. throw out low correlations between certain parameters), but does not execute properly on showing where or why this modified CMA-ES variant is needed (outside of just BBOB benchmarking). I would suggest that the authors think strongly about this question, and strengthen the motivation more.

---

### Official Review · Reviewer_6N3W · 2021-11-02

**Correctness:** 3
**Technical Novelty And Significance:** 2
**Empirical Novelty And Significance:** 2
**Recommendation:** 3
**Confidence:** 3

**Main Review:**

**Concerns on the main "block"-able assumptions**

DiBB built on the assumptions that the optimization variables (e.g., the weight and bias of neural networks in the context of neural network training problems) can be separated, and the authors assume the blocks can be constructed when it comes to the neural network (most likely feed-forward style only) each weight parameters of layers can well serve as the blocks.

However, it usually is not true when we consider the gigantic, deep, modern NNs — e.g., residual connections are designed to allow the gradient flow from the beginning of the network to the end of it. This, I think, highly violates the assumption that the authors made. In this view, I think the (proper) use of DiBB can be restricted to the quite conventional Feed forward typed neural network.

From the reported experimental results, I can conclude that DiBB works well only when the underlying objective functions are separable. It is quite natural due to the design of DiBB.


Comments about the experiments
- **COCO/BOBO experiments**: It seems like more experiments are required to understand the behavior or performance of the proposed DiBB fully.
    - DiBB shows comparable results to the CMA-ES for f1-f4 on COCO/BOBO. I think this outcome should be actual by the design of DiBB and the functions (f1-f5).
    - **Timings** sections highlight the computational times of the DiBB algorithm but do not explain the attained quality of objective values. It might be more comprehendible if it reports the objective values and computational time altogether.
- **PyBullet WALKER 2D experiments:** Similar to the COCO/BOBO experiments, the conducted experiments do not fully explain the strength or weakness of DiBB.
    - The network architecture (2 layered MLP) that has been tested may be too small to showcase the proposed algorithm. Furthermore, the reported performance of trained policy is far from the state-of-the-art or even okayish ones. One of the "classic" RL algorithms, TRPO [1] reports better performance with a shorter number of computational times.
- Minor comments:
    - Some parts of the paper use notations that are not introduced in the text. (e.g., the variable $a$ is not explained but used to explain a part of DiBB)
    - The baseline algorithms of Figure 2 are not explained.
    - Maybe some contents of table 1 are missing. However, the related contents can be found from the main text — The 2nd paragraph of **Timings**

[1] Schulman, John, et al. "Trust region policy optimization." *International conference on machine learning*. PMLR, 2015.

**Summary Of The Paper:**

This paper suggests Distributing Black-Box Optimization (DiBB) framework that enables the running of black-box optimization techniques in a distributed manner. Under the assumption that some optimization variables are correlated in a negligible manner to the optimization objective values, the authors partition the variables into the chunk of variables dubbed "Box." The core idea of DiBB is to perform black-box optimizations for each block and update the optimization results in an asynchronous manner. From the numerical experiments on the black box optimization benchmark (BBOB), the proposed DiBB shows marginally better performance when the underlying black box function matches the assumption that the optimization variables can form blocks. Additionally, the authors apply DiBB to perform a canonical example of RL tasks.

**Summary Of The Review:**

The proposed method is not techincally novel enought and the experiment result does not fully support the effectiveness of the proposed method.

---

### Official Review · Reviewer_wFNK · 2021-11-03

**Correctness:** 4
**Technical Novelty And Significance:** 3
**Empirical Novelty And Significance:** 2
**Recommendation:** 6
**Confidence:** 3

**Main Review:**

The idea of clustering variables into blocks is novel and neat. My main concern is that the empirical results are not as good as expected. However, I would still lean towards acceptance given the novelty of the idea. See detailed comments as follows.

Strengths:
1) The DiBB problem is a fundamental problem. The methodology and the ideas proposed in this paper could potentially be used in a wide range of problems of this nature, especially the neuroevolution problems.

2) The idea of clustering variables into blocks is novel and neat. Moreover, the authors have done a great job illustrating the complexities of the previous methods, including the two extremes of CMA-ES (full covariance matrix update) and sepCMA (diagonal covariance matrix update). I like the discussion of "sample complexity" and "internal complexity".

3) The line of existing works are well discussed.

4) The neuroevolution example is a good motivation.

5) In the specific method proposed, the reference solution seems to be a reasonable trick.


Weaknesses: My main concern, as state above, is that the experimental results are not as good as expected. Also, there are some clarity issues that may need to addressed.

1) In Figures 2 and 4, from the caption of the figures, it seems each figure shows results for two experiments? It is a bit hard to interpret what exactly each figure is about. For example, what do the lines ooutside the box mean?

2) In Figure 4, for the f1-f5 functions, sometimes DiBB is worse than sepCMA. This is somehow expected, but it also shows (together with the other results such as the CMA-ES results) that DiBB does not have much advantage compared with the two extremes when there is no prior knowledge about the variable dependencies. This is a major limitation of DiBB.

3) It is not clear to me how the different blocks are generated in the BBOB and BBOB large-scale suite experiments. Is it random, or by some rule?

4) In the Walker 2D example, it is not clear to me, is DiBB used in each policy update step of the RL algorithm? -- so basically, you replaced the gradient step with the DiBB update -- is that correct? If this is correct, I am wondering why don't you run a simpler task, e.g., a supervised learning task where you need to do architecture/hyperparameter search.

5) Another concern about the Walker 2D example is that, DiBB is not compared with sepCMA or CMA-ES. Theoretically, DiBB should show a large advantage in the neuroevolution task, but the baseline here is a rather weak one (Random weight guessing). This poses questions on the practical usefulness of the proposed method.

**Summary Of The Paper:**

This paper addresses the distributed black-box optimization problem. The key contribution is an in-depth understanding of the weaknesses of the current ES literature in sample complexity and per-sample complexity, and points out a direction to mitigate the weakness -- i.e., to exploit the prior knowledge of the dependency of the variables, and cluster them into blocks, where each block is solved separately with the designed distributed solution framework.

**Summary Of The Review:**

Overall, the idea is novel and neat, and the discussion of the complexity/challenge and motivation are well presented. My main concern is about the experimental results, as they are not as good as expected, especially the Walker 2D example.

It might be because of the implementation details or other reasons, but the idea seems promising.

Therefore, my overall recommendation is towards accept.

---

### Official Review · Reviewer_mpL3 · 2021-11-06

**Correctness:** 2
**Technical Novelty And Significance:** 2
**Empirical Novelty And Significance:** 2
**Recommendation:** 3
**Confidence:** 4

**Main Review:**

First of all, the paper is not very well written or sufficiently detailed to warrant a publication in such a reputed conference as this one. In addition, I think the content of the paper is more meaningful to the narrower domains focussing on evolutionary computing, such as the GECCO community. Coming to the technical side of the paper, please consider the following comments:

1) Although the abstract starts with a very ambitious note that the distributed computing framework can encapsulate "any" black-box optimizer, I do not see many provisions for simpler and very often better-performing algorithms like the Differential Evolution (DE).

2) The main framework of DiBB is quite overlapping to that of the Block-Diagonal Natural Evolution Strategies (in the paper, referred to as: Cuccu & Gomez 2012). The tweakings introduced in the paper seem minor and hence, the major novel algorithmic contributions seem limited.

3) When it comes to testing on BBOB test suite and neural architecture search problems, why the method was not compared to several other and recent distributed/parallel evolutionary algorithms where the core optimizer is not from the ES family? For example, see the following papers:

Noor Awad, Neeratyoy Mallik, and Frank Hutter, Differential Evolution for Neural Architecture Search, NAS@ICLR 2020

Z. Yang et al., "CARS: Continuous Evolution for Efficient Neural Architecture Search," 2020 IEEE/CVF Conference on Computer Vision and Pattern Recognition (CVPR), 2020, pp. 1826-1835, doi: 10.1109/CVPR42600.2020.00190.

Also, for NAS, the proposal was not compared against the most representative non-evolutionary approaches. The current results, on a first look, does not appear sufficiently competitive.

4) The paper, except for citing a reference to Omidvar et al's IEEE  TEVC paper, does not elaborate on how the linkage between different variables could be explored more thoroughly.

5) The speed-up with the proposed parallelization scheme has not been explicitly compared to other similar schemes for evolutionary/swarm intelligence-based algorithms.

6) The results were not explicitly validated on the basis of powerful, non-parametric hypothesis testing.


**Summary Of The Paper:**

The paper presents an alternative method of parallelizing a particular flavor of Evolutionary Algorithms, more specifically the Evolution Strategies (ESs) and particularly the CMA-ES. The authors propose to use a block structure to add covariance matrix information inside the blocks of potentially highly correlated variables with a low computational cost.

**Summary Of The Review:**

Considering the lack of originality, inadequate match to the theme of this conference and lack of rigorousness in experimental procedures, I will have to recommend a rejection for this paper.

---

### Author Response · Authors · 2021-11-23
**Thanks to all reviewers**

We would like to thank all Reviewers and Chairs for their effort and contribution. Four reviews constitute significant work for the community, which we truly appreciate. We decided to take heed of the comments: we will improve our work over time and resubmit in the near future. As this is OpenReview however, we would like to leave a few pointers that evidently failed to come across through our writing.

- DiBB is not in a conceptual stage, it is a realized product, *ready for public release*. We just held up the release for the sake of anonymity. It is as easy to install and use as e.g. the python package of CMA-ES.
- DiBB creates a partial-correlation abstraction of *any black-box optimization algorithm*, even methods that do not maintain a covariance matrix. It is a much broader generalization of the inspirational BD-NES, which is instead a single algorithm tied to the NES family of Evolution Strategies. DiBB can be used to distribute Simulated Annealing as well as a niching Ant Colony, if the user so desired, with minimal (and easy) setup.
- DiBB *enables evolving arbitrarily deep networks*. The target of scalable neuroevolution has been decades in the making, as it is applicable in contexts where a supervised learning interpretation (hence Deep Learning) is hard or inapplicable. The only thing limiting the progress of this research avenue has been the performance of the (most sophisticated, SOTA) search algorithm, which is now fully addressed by DiBB through distribution and parallelization. Multiple works in the literature have already highlighted the staggering difference in performance and overhead between direct policy search methods and classic deep RL approaches: and it is now possible to scale *any* neuroevolution approach to large controllers. The amount of work to explore the consequences is tantalizing, and will take years to explore. The Mujoco environment is nothing special, just a hard example application for a quick proof of concept; the true potential and scalability of DiBB should be seen in the COCO/BBOB results, and in the time scaling.

Thank you all once again for the constructive comments and useful citations contributed, we look forward to submitting more clear and appreciable results at the next opportunity.

---

### Decision · Program_Chairs · 2022-01-20

**Decision:**

Reject

**Comment:**

The authors propose a novel framework for Distributing Black-Box Optimization (DiBB) which can encapsulate any Black Box Optimization (BBO) method. DiBB overcomes some of the limitations of existing methods by leveraging expert knowledge in the problem. The reviewers raised a variety of important technical concerns. The authors seem to agree that they need to substantially rewrite the paper. Therefore I recommend a rejection.